# Environmental Factors Associated with Malocclusion in Children Population from Mining Areas, Western Romania

**DOI:** 10.3390/ijerph16183383

**Published:** 2019-09-12

**Authors:** Bianca Ioana Todor, Ioana Scrobota, Liana Todor, Alexandra Ioana Lucan, Luminita Ligia Vaida

**Affiliations:** Faculty of Medicine and Pharmacy, Department of Dentistry, University of Oradea, 410087 Oradea, Romania; biancaioana.todor@gmail.com (B.I.T.); lianatodor@rdslink.ro (L.T.); lucanalexandra@yahoo.ro (A.I.L.); ligia_vaida@yahoo.com (L.L.V.)

**Keywords:** malocclusion, ethnicity, Romani, prevalence, environment

## Abstract

Our study is based on the hypothesis that the prevalence of malocclusions in children is higher in the mining areas from North-Western (NW) Romania than in other geographic areas. We also considered that the distribution of the different types of malocclusions can be correlated with environmental factors. Therefore, the main purpose of the current study was to assess the prevalence of malocclusions in children from the mining areas in NW Romania. Another purpose was to establish the influence of certain environmental factors such as gender, geographical area of origin, and ethnicity on the distribution of malocclusions in order to provide an epidemiological reference for the planning of preventive and treatment programs adapted to the particularity of the mining areas. This cross-sectional study was performed in 2015–2016. The study batch consisted of 960 children from the mining areas, aged 7–14 years, in the period of mixed dentition and early permanent dentition. The clinical examination was conducted by a single examiner, an orthodontic specialist (TBI), in order to avoid inter-operator bias. Occlusion was registered according to Bjoerk. Occlusal clinical signs were followed for the determination of malocclusions. Most children had malocclusions (93.5%). The percentage of anomalies was significantly higher in subjects from Rosia Montana, in girls, and in the Romanians. Data showed that Angle Class I was the most prevalent malocclusion (60.21%), followed by crowding (47.5%), midline shift (43.33%), and deep bite (28.65%). The independent association between ethnicity and total malocclusions shows that the Romanian subjects presented a 3.31 higher chance of developing malocclusions than the Romani ones. The presence of malocclusions was independently influenced by all the studied environmental factors, namely gender, geographical area, and ethnicity. Our results could be relevant for oral health policy-making, i.e., planning preventive and treatment measures of malocclusions, adapted to the peculiarity of the studied mining areas.

## 1. Introduction

A malocclusion is defined as an irregularity of the teeth or a malrelationship of the dental arches beyond the range of what is accepted as normal [1,2]. Some authors consider that malocclusion is a manifestation of normal biological variability. This is a continuum ranging from an ideal occlusion to a considerable deviation from normal [3].

The World Health Organization included malocclusion under the heading of Handicapping Dento Facial Anomaly, defined as an anomaly which causes disfigurement, or which impedes function, and requiring treatment [4].

When a physical abnormality is present, especially when it affects patients’ facial aesthetics, this may have both psychological and social consequences, being in most cases a major source of stress for patients and their family. Therefore, it is important to find out the prevalence of various malocclusions and the proper methods to prevent and correct them [5,6,7].

Malocclusion could be considered as a multifactorial problem; therefore, a multitude of etiological factors have been incriminated, of which the genetic, environmental, and ethnic factors contribute the most in the emergence of malocclusions [8].

Epidemiological studies related to malocclusions have a significant contribution in orthodontic treatment planning as well as in providing a legitimate finding tool for identifying the environmental and hereditary elements in the etiology of malocclusions [9,10].

Data from epidemiological studies investigating the prevalence and severity of malocclusions in children are of great relevance to the implementation of public health programs that address orthodontic prevention and early diagnosis of malocclusions, thus contributing to the limitation of further aggravations of the malocclusions during the later stages of growth [11,12].

Numerous studies have been published regarding the prevalence of malocclusion in various populations, with the reported prevalence ranging from 39% to 98%. This variation can depend on differences in the registration methods and on differences between studied groups, regarding number, age, and ethnicity [13].

As for Europe, most of the studies regarding malocclusions were focused on Northern and Western Europe. There is very limited information available regarding the prevalence of malocclusions and orthodontic treatment needed in the Central European population. These data reported a malocclusion prevalence varying from 62.3% in Bulgaria to 70.4% in Hungary [14,15,16].

Considering Romania, there are very few studies regarding the prevalence of malocclusions. Moreover, the existing studies focused on small groups from urban areas, mainly from Central and Eastern Romania; such studies estimated a prevalence of malocclusions ranging from 50% to 75% [17].

Therefore, considering rural Romania, namely isolated and socio-environmentally impacted areas with limited access to dental care services, the epidemiological data on the oral health of such population, especially children, are missing. Such areas are former mining areas from Romania, situated in the so-called Golden Quadrilateral, Apuseni Mountains, NW Romania.

The research area selected for the current study is the Roşia Montană mining area, Apuseni Mountains, NW Romania. Gold mining was a human activity practiced almost 1900 years in the area of Roşia Montană, with environmental and socio-economic consequences. The mining activity was abandoned in 2006 with no intervention for environmental reconstruction ever since.

A health study conducted between December 2005 and January 2006 concluded that the health of the population in the Roşia Montană mining area was worse than that of the populations in the neighboring localities and that the population from Roşia Montană presented a higher frequency of chronic and acute diseases. According to the study, one of the main causes was the exposure of the population to underground and surface water pollution. The quality of the underground and surface drinking water sources was affected by the presence of several heavy metals such as iron, zinc, copper, arsenic, cadmium, mercury, selenium, and lead [18].

Ethnical diversity is an important feature of this area. The presence of the Romani population among the child population from the studied mining area is significant. The Romani population represents the largest ethnic minority in Europe, representing 1.35% of the total population in Europe. In Romania, the Romani population represents approximately 3.3% of the total population, and in the area of Roşia Montană, it represents around 12.39% [19].

The Romani population is the most populous marginalized community in Europe and has some of the greatest health needs [20]. Studies in the available literature, which refer to the state of health of children of Romani ethnicity, are very few, and those concerning the state of oral health of these children are missing. There is a need for research into the health of the Romani people, with particular emphasis on non-communicable diseases, and there is also a need for interventions that improve the Romani health [21].

Due to the scarcity of the available data regarding the prevalence and distribution of malocclusions in former mining areas, we decided to extend our study area with another mining area, namely Băița-Nucet, NW Romania.

Until the late 1960s, uranium used to be mined in the underground mines from this area. Abandoned mining activities left behind surface and underground polluted areas and poor living standards among the population [22].

The health of the population is significantly affected by environmental pollution with radon, so mortality and lung cancer rates are higher than in other regions and life expectancy is reduced by a few years [23]. Some studies suggest that the risks of illness due to heavy metals are higher for children than for adults [24].

Our hypothesis was that the prevalence of malocclusions in children is higher in the above-mentioned mining areas than in other geographic areas. We also considered that the distribution of the different types of malocclusions can be correlated with environmental factors, namely gender, geographical area and ethnicity.

Therefore, the main purpose of the current study was to assess the prevalence of malocclusions in the children from the mining areas in NW Romania.

Another purpose was to establish the influence of certain environmental factors such as gender, geographical area of origin, and ethnicity on the distribution of malocclusions in order to provide an epidemiological reference for the planning of preventive and treatment programs adapted to the particularity of the mining areas.

## 2. Materials and Methods

The studied mining zones are situated in two neighboring counties from a mountain area, NW Romania (see Figure 1). The Roşia Montană area belongs to Alba county and comprises a total population of 9640, of which 1716 are children aged 0–15 years [19]. Băița-Nucet is a smaller area, situated in Bihor county, with total population of 2165, of which 354 are children aged 0–15 years [19].

This is an epidemiological cross-sectional study that was performed in 2015–2016. The study batch consisted of 960 children (518 boys and 442 girls), aged 7–14 years, selected from a total of 1321 school-age children enrolled in three schools in the Roşia Montană area and in one school in the Băița-Nucet area. The children were in the period of mixed dentition and early permanent dentition. This batch represents 46.38% of the total age category in the studied areas (2070 children aged 0 to 15 years), considering the data of the last available population census [7].

By simple random design, the minimum sample of 714 subjects was calculated with an estimated prevalence of malocclusions of 75%, tolerable error of 3%, and 95% confidence interval. In order to compensate for a possible effect of nonresponse, the sample was increased by 30% (design effect = 1.3), totaling 952 schoolchildren [25].

Therefore, we consider that the group of 960 subjects is a significant sample from the perspective of our research.

The selected children met the inclusion criteria of being in the defined age range and in mixed or early permanent dentition. Only the consenting subjects from the selected localities were included in the study, and none of them had a history of any kind of orthodontic treatment.

Since this survey was school-based, it was impossible to obtain the treatment records from the children who had a history of orthodontic treatment; therefore, we excluded them as we were not able to determine their original occlusal status, which had already been changed. This fact may have caused some representativeness bias. [13]

However, very few children (namely 27 out of the total 1321) benefited from the orthodontic treatment, so the effects of the exclusion may be limited.

The study was conducted in accordance to the World Medical Association (WMA) Declaration of Helsinki—Ethical Principles for Medical Research Involving Human Subjects approved by the Ethics Committee of the University of Oradea, Romania. Permission to work with schoolchildren was obtained from the municipalities of Roşia Montană and Băița-Nucet as well as from the school administrations.

The children recruited for the current study were examined in the medical facilities of the studied areas or in a classroom provided by the school where they studied, according to the WHO recommendations [12]. The anamnestic data as well as the data resulting from the clinical consultation were recorded in an evaluation sheet specially designed to meet the requirements of this study. Oral health education sessions were provided for all participants.

The clinical examination was conducted by a single examiner, an orthodontic specialist (TBI), in order to avoid inter-operator bias.

Occlusion was registered according to Bjoerk [26].

Malocclusions were determined by the following clinical signs.

### 2.1. Occlusal Clinical Signs (Between the Two Jaws)

At the lateral dental group:Sagittal plan:○The presence of neutral report at 6-year molars revealing Angle Class I; neutral (normal) report was recorded when the mesiobuccal cusp of the maxillary first permanent molar occluded with the mesiobuccal groove of the mandibular first permanent molar;○The presence of mesial molar occlusion in 6-year molars, for determining Angle Class III; the mesial molar occlusion was recorded when there was deviation of at least one-half cusp width mesially to Class I;○The presence of distal molar occlusion in 6-year molars, for determining Angle Class II, was recorded when there was deviation of at least one-half cusp width distally to Class I.
Transversal plan:○The narrowing of the maxillary arch, an indicator for posterior crossbite; posterior crossbite was registered when the buccal cusps of the maxillary premolars and/or molars occluded with the lingual cusps of the opposing mandibular teeth;○The narrowing of the mandibular arch, an indicator for scissors bite; scissors bite was registered when any of the maxillary premolars and/or molars totally occluded to the buccal surface of the mandibular antagonists.
Vertical plan:○Vertical inocclusion space, an indicator for posterior open bite.

At the incisal level
Sagittal plan:○Positive overjet was recorded if the upper incisors were ahead/in front of the lower incisors;○Negative overjet was recorded when the upper incisors were behind the lower incisors.

Overjet is defined as the horizontal overlap of the incisor teeth; the overjet was measured using a graduated periodontal probe and a ruler, namely: the distance between the most labial point of the incisal edge of the upper central incisors and the labial surface of the lower central incisors, parallel to the occlusal plane; a positive overjet larger than 2 mm was considered pathologic.
Transversal plan:○Midline shift was defined as non-coincident upper and lower midlines when the posterior teeth were in contact.Vertical plan:○Overbite was considered as the vertical overlap of the incisors when the posterior teeth were in contact. An overbite exceeding 1/3 was an indicator for deep bite;○Vertical inocclusion space between the incisal edge of the maxillary central incisors and the incisal edge of the corresponding mandibular incisors when the posterior teeth were in contact was an indicator for anterior open bite.


### 2.2. Space Discrepancies (Inside One Jaw)


○Crowding was recorded when the total sum of slipped contacts, for the incisor segment, was at least 2 mm;○Spacing was recorded when the total spacing was at least 2 mm for the incisor segment.


It was considered as:Sagittal malocclusions:○Angle Class I malocclusion, the occlusion with the following clinical signs: neutral molar report, the anomaly being present at the incisal level;○Angle Class II/1 malocclusion, the occlusion with the following clinical signs: distal molar occlusion and positive overjet at the incisal level;○Angle Class II/2 malocclusion, the occlusion with the following clinical signs: distal molar occlusion, without overjet at the incisal level;○Angle Class III malocclusion, the occlusion with following clinical signs: mesial molar occlusion with or without negative overjet at incisal level.Transversal malocclusions: posterior crossbite, scissors bite, midline shift;Vertical malocclusions: posterior open bite, anterior open bite, deep bite;Space discrepancies: crowding, spacing;

The schoolchildren who presented at least one of these conditions were classified with malocclusion.

Examiner reliability for the clinical parameters was validated through intra-examiner replicate examinations for 70 subjects, using Cohen’s weighted Kappa statistics.

MedCalc Statistical Software version 19.0.3 (MedCalc Software bvba, Ostend, Belgium; https://www.medcalc.org; 2019) was used for the statistical analysis. The age of the subjects was characterized by means and standard deviation, and the nominal variables (gender, ethnicity, malocclusions) were expressed by frequency and percentage. The Chi-square test was used to determine the distribution of the different types of malocclusions by gender, geographical area, and ethnicity. In order to determine the independent association between variables (gender, geographical area, ethnicity) and various malocclusions, we used the binomial logistic regression for the multivariate analysis. Variables that reached statistical significance in the univariate analysis were introduced in the logistics regression. The odds ratio (OR) and confidence interval of 95% were calculated. A value of *p* < 0.05 was considered statistically significant.

### 2.3. Ethics Approval and Consent to Participate

The parameters of this study were approved by the Ethics Committee of the University of Oradea, Romania. The clinical examination was conducted in accordance with the World Medical Association (WMA) Declaration of Helsinki—Ethical Principles for Medical Research Involving Human Subjects. All subjects in the study had their parents’ written consent.

## 3. Results

Duplicate clinical examinations gave the following Kappa statistics: 1 for Angle Classes, open bite, and posterior crossbite; scissors bite had 0.98; crowding had 0.91; midline shift 0.82; deep bite 0.8; and spacing 0.78. These figures indicate very good intra-examiner result [27].

### 3.1. Batch Distribution

Table 1 presents the demographic data of the study group. As such, the mean age of the subjects was 10.1 years old and gender distribution was rather equal. Most of the children came from the Roşia Montană mining area and represented the majority/ethnic Romanians.

Boys and girls were equally distributed within the two mining areas, without statistical significance (*p* = 0.9). A statistically significant higher presence of Romani children was measured in the Roşia Montană area versus the Băița-Nucet area (*p* = 0.001).

### 3.2. Prevalence of Malocclusions—Overall Findings

Most children had malocclusion (93.5%). Higher statistically significant malocclusion was measured in children from the Roşia Montană mining area, girls, and Romanians. (Table 2).

Data showed that Angle Class I was the most prevalent malocclusion (60.21%), followed by crowding (47.5%), midline shift (43.33%), and deep bite (28.65%) (See Figure 2).

### 3.3. Distribution of Malocclusion Types by Environmental Factors

The distribution of malocclusion types according to geographical region, gender, and ethnicity shows that the subjects in the Roşia Montană mining area had a significantly higher frequency in crowding, midline shift, or Angle Class I, II/2, and III malocclusions.

The frequency of open bite, deep bite, and Angle Class II/1 and II/2 malocclusions was significantly higher in girls. The presence of posterior crossbite, Angle Class I, and Angle Class III malocclusions was statistically significantly higher in boys.

The presence of crowding, spacing and deep bite was significantly more frequent in the Romanian subjects, while the presence of the open bite was significantly more frequent in the Romani subjects (see Table 3).

To determine the independent association of environmental factors with the type of malocclusion, we created several models using binary logistic regression. Consequently, the model included the variables that were statistically significant by univariate analysis. Thus, the presence of malocclusion was statistically significant and independently influenced by geographical area, ethnicity. and gender (Table 4).

The independent association of environmental factors with different types of malocclusions shows that crowding or midline shift malocclusions were independently correlated with geographical area and ethnicity. Male subjects presented deep bite malocclusion less frequently. Angle Class II was independently correlated with geographical area and male gender (Table 5).

## 4. Discussions

This is the first study in Romania on the prevalence of malocclusions in a child population from mining areas, in connection with environmental factors. It is also the first study in Romania to analyze the influence of the ethnic factor on the distribution of malocclusions.

Bjoerk considers the elaboration of comparative studies on the prevalence of malocclusions in different ethnical groups considering an objective basis to be highly important, with the merits that such comparative studies could provide valuable information in understanding the causes of such anomalies [26].

The results of the epidemiological studies suggest that even those studies conducted on the same population may show great variability [13].

The comparison of the results of this study with the results of similar studies, from Romania or from other countries, may not have the necessary accuracy due to different diagnostic methods and the variation of the study batches. Furthermore, no dental cast or dental radiography was performed as part of the current study. The probability of overestimating or undervaluing certain values of the prevalence of anomalies should not be overlooked.

### 4.1. Overall Findings

The current study determined a total malocclusions prevalence of 93.5%.

Research conducted in Europe, in mixed dentition, indicates a variable prevalence of malocclusions as follows: 40% in Germany, 70.4% in Hungary, 78% in Norway, 73,8% in Albania, and 84.71% in Lithuania [13,14,28,29].

According to NHANES III, in the U.S., the prevalence of malocclusions in mixed dentation is between 20% and 55% depending on the type of anomaly. In Africa, studies on the prevalence of malocclusions in children showed a prevalence ranging from 63.8% in Tanzania to 95.6% in Libya [2,30,31,32]. In Asia, the prevalence of malocclusions ranged from 87.79% in India to 77.1% in Iran [3,33]. A study on the Latin minority in the U.S. identified a prevalence of malocclusions of 93% [34].

The comparison with the results of the above-mentioned research carried out on populations of children in Europe or other continents places the children in the studied mining areas among the children with the highest malocclusion prevalence value.

This could be explained due to environmental factors, including social factors (isolation, low standard of living, limited accessibility to specialist investigations and treatment, poor educational level).

Some authors consider that functional adaptation to environmental factors affects the surrounding structures including dentitions, bone, and soft tissue, and ultimately resulting in different malocclusion problems [8].

According to the few epidemiological studies conducted in Romania, the prevalence of malocclusions is between 41.99% and 76.9% [17,35,36].

The prevalence determined in our study is significantly higher, but the comparison is not relevant because these studies did not solely analyze the stage of mixed dentition.

Furthermore, the above-mentioned studies solely analyzed the prevalence and distribution of sagittal malocclusions, namely Angle Classes. Consequently, we may compare the results of our study with other results from similar studies exclusively from an Angle Classe perspective, as described below.

Regarding the prevalence of Angle Classes within the studied group, Angle Class I was the most common (60.21%), followed by Angle Class II/1 (21.35%), and Angle Class II/2 (13.23%), while Angle Class III was determined in the fewest cases (5.21%). These results are comparable to those reported in the Saudi population, Morocco, Lithuania, Germany, and Northeast and South of Romania [10,28,31,37,38,39,40].

Among the other types of malocclusions, crowding was the most prevalent type (47.5%), followed by midline shift (43.33%) and deep bite (28.65%). The rarest malocclusion was scissors bite (1.35%). Results are comparable with those determined in Turkish, Colombian, and German children [13,41,42].

### 4.2. The Distribution of Different Types of Malocclusion According to the Geographical Region

Based on Table 3, which shows the prevalence of different types of malocclusion according to the geographical region, the children in the Roşia Montană mining area had significantly higher frequency in crowding and midline shift than children from the Băița-Nucet mining area.

There is a correlation between those two anomalies, midline shift being a consequence of crowding. Crowding may be determined by early loss of deciduous teeth due the complications of caries, followed by early eruption of permanent teeth, in an insufficiently developed dental arch; this hypothesis has also been considered in other studies [43].

One of our previous studies on the presence of carious lesions evaluated with the Decay Missing Filling Tooth (DMFT) index that was conducted in the same mining areas determined a much higher mean DMFT value in Roşia Montană than in Băița-Nucet [44]. Thus, this could be a possible explanation for the higher presence of the above-mentioned malocclusions in Roşia Montană.

Furthermore, a significantly higher prevalence of Angle Class I, II/2, and III was determined in the Roşia Montană area. We consider that Angle Class I prevalence is directly influenced by high frequency of crowding and midline shift, in this case. Angle Class II/2 and III are malocclusions that reveal a significant genetic component, therefore one must take into consideration a genetic predisposition for these anomalies in the Roşia Montană.

On the other hand, the children from the Băița-Nucet mining area had a significantly higher prevalence of Angle Class II/1 (53.1%). This result is much higher than those reported in Romania for Angle Class II/1 at a similar age category (13.59% and 21%, respectively) [39,45].

Other studies from Turkey, Jordan, and the U.S. reported much lower Angle Class II/1 prevalence values [3,46].

Our result is comparable to the results of other studies applied on the isolated population of Hvar Island, Croatia, where the rates of inbreeding, kinship, and endogamy are high. This population showed high values of Angle Class II/1 (45.1%) [47]. According to these studies, the prevalence of the recessive genes responsible for deviations in occlusion is higher in isolated, consanguineous communities than in the rest of the population [48].

It is quite possible for the same social pattern to exist in Băița-Nucet, as both studied mining areas are isolated areas from a socio-economic point of view. In this case, the resulting genetic uniformity can cause a hereditary predisposition to this malocclusion.

The multivariate analysis confirmed these results (see Table 5), showing that the presence of malocclusion was statistically significant and independently influenced by the geographical area.

Therefore, subjects in the Băița-Nucet mining area have a lower chance of developing crowding and midline shift, and a 2.32 times higher chance of developing Angle Class II than those in the Roşia Montană mining area.

### 4.3. The Distribution of Different Types of Malocclusion According to Gender

No clear gender differences were registered, except for open bite and deep bite, both of which were more frequent in girls, and posterior crossbite, which was more frequent in boys. These data differ from those reported in a study conducted in Central Anatolia where crossbite was more often observed in females than in males, while increased overbite was more frequent in males [41].

No data on the comparison of these types of malocclusions were found in any of the studies conducted in Romania.

There were significant differences in the distribution of Angle Classes by gender: Both Angle Class II/1 and II/2 malocclusions were more frequent in girls, while Angle Class I and Angle Class III malocclusions were more frequent in boys (Table 3).

These results are both similar to those revealed by other studies conducted in Central and South Romania and unlike the results of a research on the Iranian child population, where the prevalence of Angle Class I and III malocclusions was 1.09 and 1.23 times higher, respectively, in girls than in boys. A study on the orthodontic treatment needed by Colombian children conducted by Thilander et al. also reported a higher prevalence of Class III malocclusion in girls than in boys [13,35,39,49].

Our findings show that Class II malocclusion is associated with vertical occlusal modification (open bite, deep bite) in girls, and Class III malocclusion is associated with transversal modification in boys (posterior crossbite), and this hypothesis is supported by various authors [50,51,52].

Multivariate analysis supports these results, showing that boys have lower chances of crowding, deep bite, and Angle Class II (Table 5).

### 4.4. The Distribution of Different Types of Malocclusion According to Ethnicity

The present study was conducted on two ethnicities, namely the Romanians (majority) and the Romani (minority ethnicity).

The distribution of the total malocclusions by ethnicity shows that the Romanian children presented malocclusions in 94.6% of the cases, while the Romani children presented a lower prevalence of 86.3%.

The Romanian subjects presented significantly high prevalence in crowding, midline shift, and deep bite, while the Romani subjects showed significantly frequent open bite.

We consider that the varied distribution of these types of malocclusions is due to the fact that these two ethnicities are exposed differently to certain environmental factors, namely the type of diet.

Romani children, living at the limit of subsistence, have a poor diet in sugar and carbohydrates, unlike Romanian children. The Romani children are also breastfed for a longer time, and consumption of sweetened beets is rather low [53].

As a consequence, the early loss of milk teeth, due to decay that subsequently leads to crowding and midline shift, can be lower for the Romani children.

Some authors report that prolonged breastfeeding decreases the risk of malocclusion [54,55].

Therefore, the early loss of deciduous teeth, due to decay, with the compromising of the support dental area, could be a possible explanation of the higher frequency of crowding, midline shift, and deep bite in the Romani subjects.

A study conducted in a population of children from Tanzania showed a higher prevalence of open bite in children belonging to the less privileged ethnicity. This high prevalence was associated with prolonged thumb-sucking. This may be the explanation for the much higher open bite frequency in the Romani children [2].

We have not found data for comparison regarding prevalence and distribution of malocclusions in children of Romany ethnicity, because apparently there are no studies on their oral health.

There were no statistically significant differences in the distribution of Angle Classes according to ethnicity, with the percentages being similar (Table 3).

However, we observed a predominance of Angle Class II/1 in Romani children, a result comparable to those of the studies in Pakistan, with the origin of these two populations being common [21,56,57].

Angle Class II/2 was predominant in the Romanian children, with much higher values than those reported by other studies [13,45,58].

We must admit the limitations of our findings being from a cross-sectional study; we cannot provide definitive proof about cause-and-effect relationships. Our goal was to provide information that might be useful in further research on this important topic.

If we follow the independent association of ethnicity with total malocclusions, the Romanian subjects have a 3.31 higher chance of developing malocclusions than the Romani subjects (Table 4).

As for crowding, the chance of developing this anomaly is 2.03 times higher for the Romanian subjects than for the Romani subjects.

For the midline shift, the Romanian subjects have a 1.6-times higher chance of developing this anomaly than the Romani subjects and a 3.85% higher chance of developing deep bite than the Romani subjects (Table 5).

The Romani are a marginalized minority with poor living standards and without access to dental care. Within the Romany community, malocclusion is influenced by multiple etiological environmental factors such as poor oral hygiene, poor diet, and bad oral habits. Despite this aspect, the odds of developing malocclusions are much lower than for the Romanians.

Even though a significantly higher prevalence of Romani children in the Roşia Montană mining area than in the Băița-Nucet mining area was measured (see Table 1), both ethnicity and geographical area have maintained an independent influence on malocclusion prevalence.

Further studies are necessary to understand whether the lower prevalence of malocclusions in Romani children is due to environmental factors or to other causes, such as those of genetic order. In the etiology of malocclusion, the genetic factor seems to have the greatest influence whilst environmental factors appear to be of minor importance, but the permanent interaction between heredity and environment, both of a continually modeling nature, determines the occlusal morphology in every moment of life [59].

So far no epidemiological studies regarding malocclusions prevalence in Romania have been conducted, based on a complex method of diagnosis, in order to achieve relevant comparisons. Further exhaustive research, developed in other Romanian areas, are necessary as our findings may offer valuable data for comparison.

The most important finding in this study was the identification of malocclusions, which are independently correlated with environmental factors. This information is relevant for oral health policy-making, i.e., planning preventive measures adapted to the peculiarity of the studied mining areas.

Furthermore, the high values of malocclusion prevalence measured in mixed dentition determined in our research highlight the importance of early orthodontic screening. Therefore, orthodontic prevention and interception programs should be implemented, starting with pre-school children, which requires close collaboration between schools, parents, orthodontists, and pediatricians.

## 5. Conclusions

The current study demonstrates that malocclusions are an important health problem in the mining areas from NW Romania, sustained by a very high prevalence of 93.5%.

There was an independent association between malocclusions and all the studied environmental factors, namely gender, geographical area, and ethnicity.

Our results could be relevant for oral health policy-making, i.e., planning preventive and treatment measures of malocclusions adapted to the peculiarity of the studied mining areas.

## Figures and Tables

**Figure 1 ijerph-16-03383-f001:**
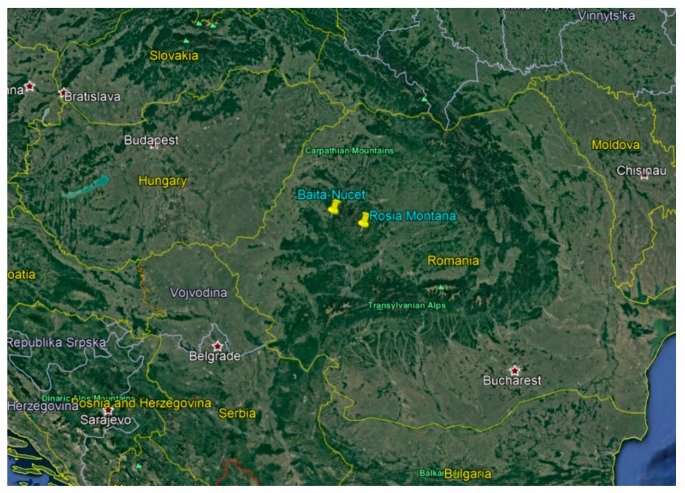
Geographical location of the two studied mining areas (source—Google maps).

**Figure 2 ijerph-16-03383-f002:**
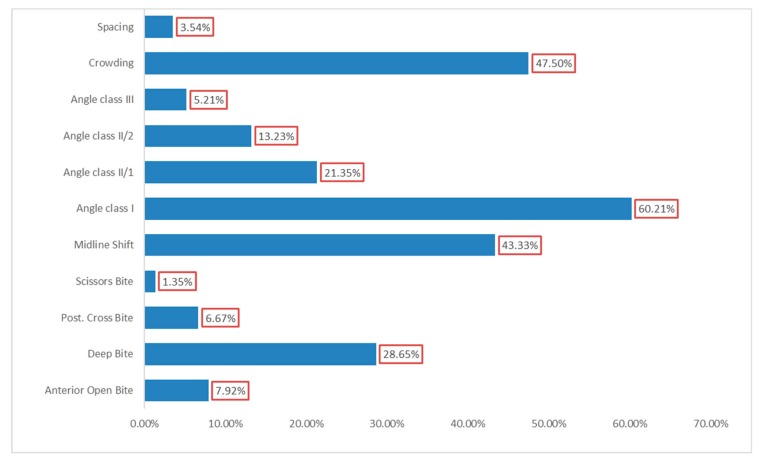
Prevalence of different types of malocclusion.

**Table 1 ijerph-16-03383-t001:** Demographic data of subjects within the study.

Variables				Results
Age (years)		Roşia Montană mining area	Băița-Nucet mining area	10.1 ± 1.9
Gender	Boys	391 (46.2%)	51 (45.1%)	442 (46%)
	Girls	456 (53.8%)	62 (54.9%)	518 (54%)
Ethnicity	Romanian	726 (85.7%)	110 (97.3%)	836 (87.1%)
	Romani	121 (14.3%)	3 (2.7%)	124 (12.9%)
Geographical area	RM mining area	847 (88.2%)	n/a	847 (88.2%)
	B-N mining area	n/a	113 (11.8%)	113 (11.8%)

**Table 2 ijerph-16-03383-t002:** Distribution of total malocclusions by environment, gender and ethnicity.

Variables	Values	*p*-Value
Total malocclusion	898 (93.5%)	<0.0001
Without malocclusion	62 (6.5%)	
Roşia Montană mining area	798 (94.2%)	0.03
Băița-Nucet mining area	100 (88.5%)	
Girls	421 (95.2%)	0.04
Boys	477 (92.1%)	
Romani	107 (86.3%)	0.001
Romanians	791 (94.6%)	

**Table 3 ijerph-16-03383-t003:** Distribution of malocclusions by geographical region, gender and ethnicity.

Variables		Roşia Montană Mining Area	Băița-Nucet Mining Area	*p*	Girls	Boys	*p*	Romani/Minority	Romanians/Majority	*p*
**Space Anomalies**										
Crowding		434 (51.2%)	22 (19.5%)	<0.001	220 (49.8%)	236 (45.6%)	0.2	44 (35.5%)	412 (49.3%)	0.006
Spacing		25 (3.0%)	9 (8.0%)	0.1	14 (3.2%)	20 (3.9%)	0.6	-	34 (4.1)	0.01
**Vertical Plan**										
Open Bite *		70 (8.3%)	6 (5.3%)	0.4	40 (9%)	36 (6.9%)	0.007	25 (20.2%)	51 (6.1%)	<0.001
Deep Bite		244 (28.8%)	31 (27.4%)		145 (32.8%)	130 (25.1%)		31 (25%)	244 (29.2%)	
**Transversal Plan**										
Post. Crossbite		55 (6.5%)	9 (8%)	0.3	24 (37.5%)	40 (62.5%)	0.004	4 (3.2%)	60 (7.2%)	0.1
Scissors Bite		13 (1.5%)	-	0.1	9 (0.2%)	7 (0.1%)	0.6	2 (0.1%)	14 (0.1%)	1
Midline Shift		394 (46.5%)	22 (19.5%)	<0.001	192 (43.4%)	224 (43.2%)	1.000	44 (35.5%)	372 (44.5%)	0.07
**Sagital Plan**										
Angle Class	I	531 (62.7%)	47 (41.6%)	<0.001	243 (55%)	335 (64.7%)	<0.001	78 (62.9%)	500 (59.8%)	0.2
	II/1	145 (17.1%)	60 (53.1%)		120 (27.1%)	85 (16.4%)		31 (25%)	174 (20.8%)	
	II/2	123 (14.5%)	4 (3.5%)		65 (14.7%)	62 (12%)		11 (8.9%)	116 (13.9%)	
	III	48 (5.7%)	2 (1.8%)		14 (3.2%)	36 (6.9%)		4 (3.2%)	46 (5.5%)	
Angle Class **	I	531 (62.7%)	47 (41.6%)	<0.001	243 (55%)	335 (64.7%)	<0.001	78 (62.9%)	500 (59.8%)	0.5
	II–III	316 (37.3%)	66 (58.4%)		199 (45%)	183 (35.3%)		46 (37.1%)	336 (40.2%)	

“*”—As there were only four cases of posterior open bite, all of them measured in Romanian subjects, we summed these cases with anterior open bite cases in one category, namely “open bite”. “**”—We split the Angle Classes into two categories by molar report (normal vs. pathological) for multivariate analyses.

**Table 4 ijerph-16-03383-t004:** Multivariate analysis for total malocclusions.

Variables	B	*p*	OR	95% CI	
**Malocclusion**					
Băița-Nucet mining area	−0.95	0.005	0.38	0.19	0.74
Romanians/majority	1.19	<0.001	3.31	1.80	6.11
Male gender	−0.64	0.02	0.52	0.30	0.90

**Table 5 ijerph-16-03383-t005:** Multivariate analysis for types of malocclusions.

Variables	B	*P*	OR	95% CI	
**Crowding**					
N-B mining area	−1.55	<0.001	0.21	0.13	0.34
Romanians/majority	0.71	<0.001	2.03	1.36	3.03
Male gender	−0.24	0.07	0.78	0.60	1.02
**Midline Shift**					
N-B mining area	−1.32	<0.001	0.26	0.16	0.43
Romanians/majority	0.47	0.01	1.60	1.08	2.38
**Deep Bite**					
Romanians/majority	1.34	<0.001	3.85	2.08	7.13
Male gender	−0.09	0.7	0.90	0.53	1.54
**Angle Class II**					
N-B mining area	0.84	<0.001	2.32	1.55	3.47
Male gender	−0.39	0.004	0.67	0.52	0.88

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
