# Peer review of "Environmental Factors Associated with Malocclusion in Children Population from Mining Areas, Western Romania"

_ijerph, 2019, doi:10.3390/ijerph16183383_

Round 1
Reviewer 1 Report
I advise the authors to use the STROBE statement for observational studies to conduct the methodology, making it more detailed and reproducible, also explaining the details of the sample size.
Author Response
Please see the attachment.
Yours truly,
Dr. Ioana Scrobota

Reviewer 2 Report
The submitted manuscript provides insight into the potential effects that environmental factors and sociodemographic factors may have on malocclusion in children located in two mining regions within Romania.
In order to demonstrate the potential effect of geography (location in a mining region), it would have been beneficial to obtain data on a Romanian sample outside of the two mining regions for comparison. It should be clearly justified and explained why this was not done.
Gender and ethnicity, in addition to mining area, were found to be significant factors associated with malocclusion in a logistic regression model. However, gender and ethnicity are not compared between the 2 mining regions.
Numerous grammatical errors in the manuscript are noted. This manuscript would benefit from additional editing, preferably by a professional editing service or native English language speaker.
Author Response

(The authors gave the same response as above.)

Reviewer 3 Report
Very interesting paper and well designed study
Discussion section could be implemented because it’s important to underline that effective orthodontic prevention programs should therefore carried out through the School (Vozza I, et al. Clin Ter. 2019) being aware paediatricians of the importance of early first dental visit (Luzzi V, et al. J Clin Exp Dent. 2017)
Author Response

(The authors gave the same response as above.)
